# The relationship between screen time, screen content for children aged 1-3, and the risk of ADHD in preschools

Jian-Bo Wu[1], Yanni Yang[2], Qiang Zhou[1], Jiemin Li[1], Wei-Kang Yang[1], Xiaona Yin[1], Shuang-Yan Qiu[1], Jingyu Zhang[1], Minghui Meng[3], Yawei Guo[4], Jian-hui Chen[5]☯*, Zhaodi Chen🄳[6]☯

1 Department of Clinical Psychology, Shenzhen Longhua Maternity and Child Healthcare Hospital, Shenzhen, China, 2 ShenZhen PingShan XinHe Experimental School, Shenzhen, China, 3 Shenzhen Longhua District Longlan School affiliated Xintang kindergarten, Shenzhen, China, 4 School of Public Health, Sun Yat-sen University, Guangzhou, China, 5 Guangdong Second Traditional Chinese Medicine Hospital, Guangzhou, China, 6 Department of health education and promotion, Shenzhen Longhua Maternity and Child Healthcare Hospital, Shenzhen, China

☯ These authors also contributed equally to this work.
* chenjh367@mail3.sysu.edu.cn

## Abstract

### Objective

This study investigates the relationship between screen time, screen content, and the risk of Attention Deficit Hyperactivity Disorder (ADHD) using data from a large sample. Specifically, it examines how different types of screen content (such as educational videos, cartoon videos, and interactive videos) are associated with the risk of ADHD. The aim is to offer a scientific foundation for the rational management of children's screen time and screen content.

### Methods

We collected data through a questionnaire survey involving a study population of 41,494 children from Longhua District, Shenzhen City, China. The questionnaire recorded the daily screen time and the type of content viewed by the children at ages 1–3 years and assessed their risk of ADHD using the Strengths and Difficulties Questionnaire (SDQ) at ages 4–6 years. Hierarchical logistic regression analysis, controlling for confounding factors, was employed to explore the associations between screen time, screen content, and ADHD risk.

### Results

In the total sample, 6.7% of the participants had screen time exceeding 60 minutes per day, with educational videos predominant type (63.4%). 16.5% of the participants were identified as being at risk for ADHD. Statistically significant positive associations with ADHD were observed across all categories of screen time ($P<0.001$). Moreover, as screen time increased, the risk of ADHD also rose ($OR_{1\sim60\ mins/d}=1.627$, $95\%CI=1.460\sim1.813$; $OR_{61\sim120\ mins/d}=2.838$, $95\%CI=2.469\sim3.261$; $OR_{>120\ mins/d}=3.687$, $95\%CI=2.835\sim4.796$).

**Data availability statement:** All relevant data are within the paper and its Supporting Information files.

**Funding:** Funding Longhua District Medical And Health Institutions Regional Scientific Research Project (2022086). Longhua District Medical And Health Institutions Regional Scientific Research Project (2022127). Medical Key Discipline Construction Fund of Longhua District.

**Competing interests:** Disclosure statement The authors declare no potential conflict of interests.

Significant positive associations with ADHD were observed across all categories of screen time in the educational videos and cartoon videos. For the educational videos group, the odds ratios were as follows: $OR_{1-60\ mins/day}$=1.683 ($95\%\ CI$=1.481–1.913), $OR_{61-120\ mins/day}$=3.193 ($95\%\ CI$=2.658–3.835), and $OR_{>120\ mins/day}$=3.070 ($95\%\ CI$=2.017–4.673). For the cartoon videos group, the odds ratios were: $OR_{1-60\ mins/day}$=1.603 ($95\%\ CI$=1.290–1.991), $OR_{61-120\ mins/day}$=2.758 ($95\%\ CI$=2.156–3.529), and $OR_{>120\ mins/day}$=4.097 ($95\%\ CI$=2.760–6.081). However, no significant associations with ADHD risk were found for any category of screen time in the interactive videos group ($OR_{1-60\ mins/d}$=0.744, $95\%CI$=0.361~1.534; $OR_{61-120\ mins/d}$=0.680, $95\%CI$=0.296~1.560; $OR_{>120\ mins/d}$=1.678, $95\%CI$=0.593~4.748).

## Conclusion

Increased screen time is associated with a higher risk of ADHD, particularly for educational and cartoon videos, while interactive videos show no significant link. To mitigate this risk, parents and educators should implement strategies such as setting time limits, encouraging breaks, and promoting alternative activities. Future research should focus on longitudinal studies and intervention trials to further explore and address this relationship.

## Introduction

ADHD, a common neurodevelopmental disorder, is characterized by core symptoms such as inattention, hyperactivity, and impulsivity, which pose challenges to children's learning, social interaction, and emotional regulation abilities [1,2]. Although the exact pathophysiology of ADHD is not fully understood, environmental factors-including lifestyle factors-are recognized as playing a significant role in its development [3]. Among these factors, the relationship between screen time and ADHD risk has been a hot topic of research in recent years. However, studies specifically investigating the association between different type of screen content and ADHD risk remain scarce.

In recent years, the widespread adoption of electronic devices has significantly increased children's screen time, raising widespread societal concern about the impact of electronic screens on children's mental health. As research deepens, people have become aware that, in addition to the total amount of screen time, the type of screen content also has a significant impact on children's psychological development [4,5]. Most existing research focuses on the association between screen time and the risk of ADHD, highlighting its potential impact on children's mental health development [6,7]. However, these studies often overlook the fact that different types of screen content may have distinct effects on children's mental health [8].

Existing studies on the link between screen time and hyperactive behavior generally indicate that excessive screen time may indirectly increase the risk of ADHD by affecting children's sleep, physical activity, and social interaction [9,10]. However, these studies often fail to explore the differential impacts of various types of screen contents on children's mental health in depth [11,12]. For instance, educational videos, often perceived as beneficial for learning, may nevertheless contain rapidly changing visuals and intense sensory stimulation that adversely affect children's ability to concentrate and self-regulate. Similarly, cartoon videos, with their vibrant colors and exaggerated movements, may overstimulate children's attentional systems, leading to decreased attention to the real world and affecting cognitive and emotional development [13]. In contrast, interactive videos, which promote children's active participation and social interaction, may have different effects on ADHD risk, though their specific impact requires further investigation.

This study leverages the extensive data resources from the Longhua Child Cohort Study (LCCS) in Shenzhen to comprehensively assess the relationship between children's screen time, different types of screen content, and the risk of ADHD. The aim is to provide scientific evidence to guide parents and educators in reasonably managing children's screen time and optimizing screen content selection, thereby effectively reducing the risk of ADHD in children.

## Methods

### Study design and participants

This study, conducted in accordance with the Declaration of Helsinki, used census methods to source data from the 2021 survey of the LCCS, covering all kindergartens in Longhua District, Shenzhen. The LCCS was a large-scale epidemiological survey conducted in Longhua District, Shenzhen, China, aiming to assess the impact of children's lifestyle habits on early psychological and behavioral development of preschoolers. We focused on the early childhood period (1–3 years) for screen time and content data, given its pivotal role in development. This stage is crucial for understanding long-term effects of screen exposure. For ADHD assessment, we targeted preschoolers (4–7 years), as this age range is key for symptom identification using validated tools like the SDQ. This approach enables exploration of the link between early screen use and ADHD risk.

The research project was conducted across 250 kindergartens in the Longhua District of Shenzhen in 2021. From 8th October to 23th November 2021, the project was publicized to the parents of kindergarten children, encouraging their participation. After obtaining parental consent, informed consent forms were signed by the parents, and a questionnaire survey was conducted. A total of 59,600 questionnaires were distributed, and 56,740 were returned, yielding a response rate of 95.2%. After excluding 15,246 questionnaires with incomplete information, the final sample size was 41,494. This study was approved by the Biomedical Research Ethics Review Committee of the School of Public Health, Sun Yat-sen University (Ethics Approval number: 2021 No. 116).

### Data collection

The questionnaire collected information on family demographic characteristics, daily screen time, and the types of programs viewed during screen time by children when they were at the age of 1–3 years old. Additionally, it assessed the risk of ADHD using the Strengths and Difficulties Questionnaire (SDQ) for them at the age of 4–7 years old. All participants had signed the Human Ethics and Consent to Participate forms and agreed to be involved in this study.

### Measurement of screen time (major exposure variables) and category

"Screen time" was defined as time spent looking at screens such as phones, TVS, tablets or desktop computers, game consoles, as reported by the children's parents. We chose to collect information on screen time for children aged 4–7. An ordinal categorical survey was conducted to assess screen time, and a nominal categorical survey was conducted to evaluate the types of programs viewed during screen time (Table 1).

### Measurement of ADHD risk

In this study, the Strengths and Difficulties Questionnaire (SDQ) was used to assess the risk of ADHD in children. The SDQ, developed by American psychologist Goodman in 1997, is

**Table 1. Questions and options regarding the screen time and program of the screen time.**

| No. | Questions | Options |
|---|---|---|
| Q1 | How long was the screen time of your child/children at his/her/their age of 1–3 years old per day? | 0 mins/d<br>1~60 mins/d<br>61~120 mins/d<br>>120 mins/d |
| Q2 | What types of programs did your child/children primarily view during screen time at his/her/their age of 1–3 years old? | Educational videos<br>Cartoon videos<br>Interactive videos |

*Educational videos：(with explicit prosocial or cognitive components [e.g., Sesame Street and QiaoHu, a similar program in China])

*Cartoon videos：(e.g., Thomas & Friends and Super Flying Man)

*Interactive videos (e.g., apps where kids tap, swipe, or use controllers to play and family video chats real-time conversations between relatives using video calling apps).

0 mins/d: Refers to children aged 1–3 who do not have any exposure to electronic screens, or have occasional exposure but use electronic screens for less than 1 minute per day.

a concise behavioral screening scale [14]. In 2005, norms for the Chinese population were established, ensuring cultural relevance and validity [15]. The scale consists of 25 items, covering 5 dimensions: emotional symptoms, conduct problems, ADHD symptoms, peer problems, and prosocial behavior. Items on the SDQ are rated on a scale from 0 to 2, with 0 indicating no agreement, 1 indicating partial agreement, and 2 indicating perfect agreement. The total score for ADHD symptoms ranges from 0 to 5 for normal, 6 for borderline, and 7–10 for abnormal.

Based on these scores, participants can be categorized into a normal group ($\leq 5$) and an ADHD risk group ($\geq 6$). The ADHD subscale cut-offs used in this study are based on established norms and research, and these thresholds have been validated in multiple studies and are widely accepted in both research and clinical practice. Furthermore, the cultural adaptation of the SDQ for the Chinese population has undergone a rigorous validation process to ensure its accuracy and applicability in this context [16]. The scale demonstrates good reliability, with a Cronbach's α coefficient of 0.749 [17,18].

## Covariates

The following confounding covariates were included in the analysis: child's gender, age, parental marital status, parents' educational attainment, household monthly income, single-child status and explaining the content of screen time program to the child.

## Statistical analysis

Descriptive statistics were used to characterize the study population. Mean ± standard deviation (SD) and sample number (percentage) were presented for continuous and categorical variables, respectively. A chi-square test was used to compare differences in screen time, types of programs viewed during screen time, and covariate variables among ADHD risk groups. Logistic regression analysis was employed to explore the association between screen time and ADHD risk.

Based on the existing literature and theoretical considerations, we believe that interactions between covariates are unlikely to have a significant impact on the primary study outcomes. The main objective of this study is to evaluate the effect of individual covariates on prognosis. Therefore, we prioritized assessing main effects rather than interaction effects. Nonetheless, future research could further explore interactions between covariates to gain a more comprehensive understanding of how various factors influence prognosis.

## Results

### Analysis of demographic characteristics, screen time, types of screen content, and covariate variables among ADHD risk groups

Participants' SDQ scores and associated ADHD risk levels are presented in Table 2. In the total sample, we found that 7.5% of the participants exhibited abnormal levels of ADHD symptoms (defined as a score of 7–10). Additionally, 9.0% of the participants were on the borderline for ADHD symptoms (defined as a score of 6). Overall, 16.5% of the participants were at risk for ADHD.

Participants' demographics and characteristics are summarized in Table 3. A total of 41,494 children (22,113 boys [53.3%] and 19,381 girls [46.7%]; mean [SD] age, 5.13±0.67 years old) completed the questionnaire. The risk of ADHD was higher in boys compared to girls(18.9% vs. 13.7%, $^2$=201.855, $P$<0.001).

### Relationship between screen time and ADHD risk

We performed a logistic regression analysis to investigate the associations between screen time and ADHD risk (Table 4). Statistically significant positive associations with ADHD were observed across all categories of screen time ($\beta_{1\sim60\ mins/d}$ = 0.493, $\beta_{61\sim120\ mins/d}$ = 1.041, $\beta_{>120\ mins/d}$ = 1.302, $P$<0.001). Additionally, as screen time increased, the risk of ADHD also rose ($OR_{1\sim60\ mins/d}$=1.637, $95\%CI$=1.469~1.824; $OR_{61\sim120\ mins/d}$=2.833, $95\%CI$=2.466~3.255; $OR_{>120\ mins/d}$=3.676, $95\%CI$=2.827~4.778). Forest plot for the odds ratios can be seen in Fig 1.

Age, parental education attainment, household monthly income, Types of programs viewed during screen time and screen time was transferred into dummy variables. Abbreviation: $\beta$, coefficient of Logistic Regression Analysis with adjustment for age, gender, parental marital status, maternal and paternal education attainment, household monthly income, single child status, types of programs viewed during screen time and discuss the content of screen time program with the child. $OR$, odds ratio of Logistic Regression Analysis. $95\%CI$, 95% confidence interval of $OR$ of Logistic Regression Analysis. Bold font indicates statistical significance.

### Relationship between screen time and ADHD risk in different types of programs viewed during screen time

We conducted a stratified logistic regression analysis to further investigate the associations between screen time and ADHD risk across different types of programs viewed during screen time (Table 5). Statistically significant positive associations with ADHD were observed across all categories of screen time in the educational videos and cartoon videos groups. (education videos group [$OR_{1\sim60\ mins/d}$=1.683, $95\%CI$=1.481~1.913; $OR_{61\sim120\ mins/d}$=3.193, $95\%CI$=2.658~3.835; $OR_{>120\ mins/d}$=3.070, $95\%CI$=2.017~4.673]; Cartoon group [$OR_{1\sim60\ mins/d}$=1.603, $95\%CI$=1.290~1.991; $OR_{61\sim120\ mins/d}$=2.758, $95\%CI$=2.156~3.529;

**Table 2. ADHD scores and risk of the Strengths and Difficulties Questionnaire (SDQ).**

| Category | ADHD scores (Mean±SD) | No. (%) |
|---|---|---|
| Total | 3.69±1.93 | 41494 (100.0) |
| Normal | 3.09±1.45 | 34645 (83.5) |
| ADHD risk (Edge value) | 6.00±0.00 | 3737 (9.0) |
| ADHD risk (Abnormal) | 7.60±0.84 | 3112 (7.5) |

**Table3. Participants sociodemographic characteristics and differences in screen time, types of programs viewed during screen time, and covariate variables among ADHD risk groups (N=41,494).**

| Characteristics | Total | Normal group | Risk group | ² | P |
|---|---|---|---|---|---|
| **Screen time** | | | | | |
| 0 mins/d | 3777 (9.1) | 3359 (88.9) | 418 (11.1) | 337.445 | **<0.001** |
| 1~60 mins/d | 34943 (84.2) | 29276 (83.8) | 5667 (16.2) | | |
| 61~120 mins/d | 2464 (5.9) | 1804 (73.2) | 660 (26.8) | | |
| >120 mins/d | 310 (0.8) | 206 (66.5) | 104 (33.5) | | |
| **Types of programs viewed during screen time** | | | | | |
| Educational videos | 26320 (63.4) | 22524 (85.6) | 3796 (14.4) | 270.463 | **<0.001** |
| Cartoon videos | 14516 (35.0) | 11657 (80.3) | 2859 (19.7) | | |
| Interactive videos | 658 (1.6) | 464 (70.5) | 194 (29.5) | | |
| **Explaining the content of screen time program to the child** | | | | | |
| Yes | 29587(71.3) | 24997(84.5) | 4590(15.5) | 73.686 | **<0.001** |
| No | 11907(28.7) | 2259(19.0) | 9648(81.0) | | |
| **Gender** | | | | | |
| Male | 22113 (53.3) | 17927 (81.1) | 4186 (18.9) | 201.855 | **<0.001** |
| Female | 19381 (46.7) | 16718 (86.3) | 2663 (13.7) | | |
| **Age** | | | | | |
| 4~5 years old (< 5) | 18106 (43.6) | 15080 (83.3) | 3026 (16.7) | 1.368 | 0.505 |
| 5~6 years old (< 6) | 17935 (43.2) | 15018 (83.7) | 2917 (16.3) | | |
| 6~7 years old (< 7) | 5453 (13.1) | 4547 (83.4) | 906 (16.6) | | |
| **Parental marital status** | | | | | |
| Married | 39682 (95.6) | 33199 (83.7) | 6483 (16.3) | 18.747 | **<0.001** |
| Remarried/Divorced/Widowed | 1812 (4.4) | 1446 (79.8) | 366 (20.2) | | |
| **Maternal education attainment** | | | | | |
| Junior high school and below | 6408 (15.4) | 5017 (78.3) | 1391 (21.7) | 296.555 | **<0.001** |
| High school or technical secondary school | 8671 (20.9) | 6995 (80.7) | 1676 (19.3) | | |
| Junior college | 25175 (60.7) | 21502 (85.4) | 3673 (14.6) | | |
| Undergraduate and above | 1240 (3.0) | 1131 (91.2) | 109 (8.8) | | |
| **Paternal education attainment** | | | | | |
| Junior high school and below | 5953 (14.3) | 4658 (78.2) | 1295 (21.8) | 282.356 | **<0.001** |
| High school or technical secondary school | 8904 (21.5) | 7188 (80.7) | 1716 (19.3) | | |
| Junior college | 24728 (59.6) | 21081 (85.3) | 3647 (14.7) | | |
| Undergraduate and above | 1909 (4.6) | 1718 (90.0) | 191 (10.0) | | |
| **Household monthly income** | | | | | |
| <¥10,000 | 7266 (17.5) | 5760 (79.3) | 1506 (20.7) | 252.489 | **<0.001** |
| ¥10,000~20,000 | 14331 (34.5) | 11735 (81.9) | 2596 (18.1) | | |
| ¥20,000~30,000 | 8724 (21.0) | 7386 (84.7) | 1338 (15.3) | | |
| ¥30,000~40,000 | 4648 (11.2) | 4062 (87.4) | 586 (12.6) | | |
| ¥40,000 | 6525 (15.7) | 5702 (87.4) | 823 (12.6) | | |
| **Single child status** | | | | | |
| Yes | 13154 (31.7) | 10568 (80.3) | 2586 (19.7) | 138.966 | **<0.001** |
| No | 28340 (68.3) | 24077 (85.0) | 4263 (15.0) | | |

Abbreviation: ², coefficient of chi-square test. Bold font indicates statistical significance.

**Table 4. Relationship between screen time and ADHD risk (N=41,494).**

| Characteristics | $\beta$ | OR (95%CI) | P |
|---|---|---|---|
| **Age** | | | |
| 4~5 years old (< 5) | Ref. | | |
| 5~6 years old (< 6) | -0.020 | 0.981 (0.927~1.038) | 0.499 |
| 6~7 years old (< 7) | -0.042 | 0.959 (0.883~1.042) | 0.323 |
| **Gender** | -0.365 | 0.694 (0.658~0.732) | **<0.001** |
| **Parental marital status** | 0.189 | 1.207 (1.069~1.364) | **0.002** |
| **Maternal education attainment** | | | |
| Junior high school and below | Ref. | | |
| High school or technical secondary school | -0.032 | 0.968 (0.887~1.057) | 0.473 |
| Junior college | -0.224 | 0.799 (0.733~0.872) | **<0.001** |
| Undergraduate and above | -0.616 | 0.540 (0.431~0.676) | **<0.001** |
| **Paternal education attainment** | | | |
| Junior high school and below | Ref. | | |
| High school or technical secondary school | -0.067 | 0.935 (0.856~1.021) | 0.133 |
| Junior college | -0.224 | 0.799 (0.732~0.873) | **<0.001** |
| Undergraduate and above | -0.412 | 0.662 (0.552~0.795) | **<0.001** |
| **Household monthly income** | | | |
| <¥10,000 | Ref. | | |
| ¥10,000~20,000 | -0.074 | 0.929 (0.862~1.001) | 0.054 |
| ¥20,000~30,000 | -0.178 | 0.837 (0.766~0.915) | **<0.001** |
| ¥30,000~40,000 | -0.345 | 0.708 (0.634~0.792) | **<0.001** |
| ¥40,000 | -0.322 | 0.725 (0.655~0.802) | **<0.001** |
| **Types of programs viewed during screen time** | | | |
| Educational videos | Ref. | | |
| Cartoon videos | 0.288 | 1.333 (1.262~1.408) | **<0.001** |
| Interactive videos | 0.726 | 2.066 (1.733~2.463) | **<0.001** |
| **Explaining the content of screen time program to the child** | -0.265 | 0.767 (0.724~0.813) | **<0.001** |
| **Screen time** | | | |
| 0 mins/d | Ref. | | |
| 1~60 mins/d | 0.493 | 1.637 (1.469~1.824) | **<0.001** |
| 61~120 mins/d | 1.041 | 2.833 (2.466~3.255) | **<0.001** |
| >120 mins/d | 1.302 | 3.676 (2.827~4.778) | **<0.001** |

Model fit information: $\chi^2$=9.105, df=8, P=0.333.

$OR_{>120 \text{ mins/d}}$=4.097, 95%CI=2.760~6.081]) However, no category of screen time was significantly associated with ADHD risk in the interactive videos group ($OR_{1~60 \text{ mins/d}}$=0.744, 95%CI=0.361~1.534; $OR_{61~120 \text{ mins/d}}$=0.680, 95%CI=0.296~1.560; $OR_{>120 \text{ mins/d}}$=1.678, 95%CI=0.593~4.748). Distribution of children at risk of ADHD across different screen time categories can be seen in Table 6.

Screen time was transferred into dummy variables. Abbreviation: OR (95%CI), odds ratio (95% confidence interval of OR) of Logistic Regression Analysis with adjustment for age, gender, parental marital status, maternal and paternal education attainment, household monthly income, single child status and explaining the content of screen time program to the child. Bold font indicates the non-significance of interactive videos.

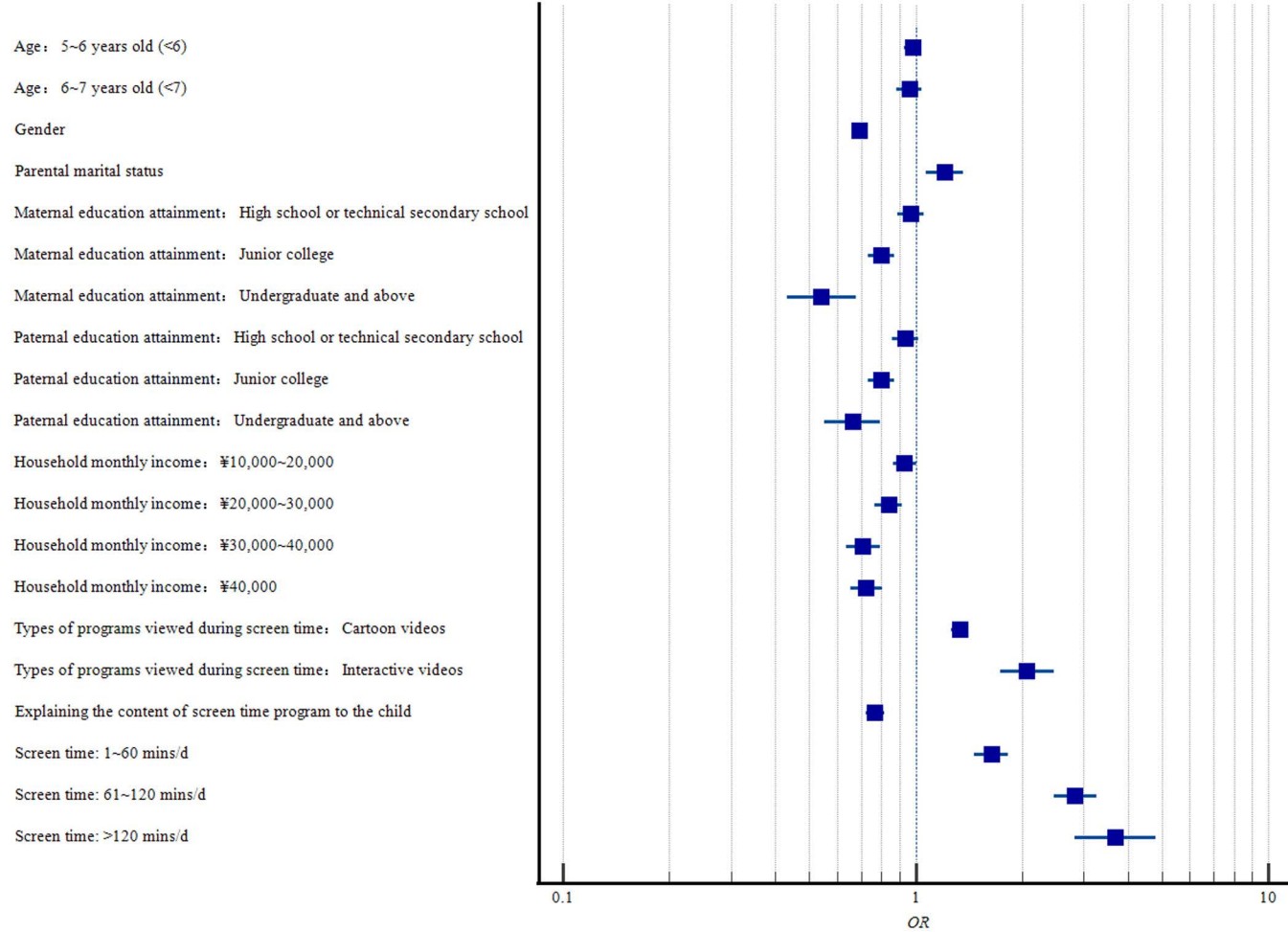

**Fig 1. Odd ratios for each covariates and screen time.**

**Table 5. Relationship between screen time and ADHD risk in different types of programs viewed during screen time (N=41,494).**

| Type of content viewed during screen time | Educational videos (n=26320) | Cartoon videos (n=14516) | Interactive videos (n=658) |
|---|---|---|---|
| 0 mins/d | Ref. | | |
| 1~60 mins/d | 1.683 (1.481~1.913) | 1.603 (1.290~1.991) | 0.744 (0.361~1.534) |
| 61~120 mins/d | 3.193 (2.658~3.835) | 2.758 (2.156~3.529) | 0.680 (0.296~1.560) |
| >120 mins/d | 3.070 (2.017~4.673) | 4.097 (2.760~6.081) | 1.678 (0.593~4.748) |

## Discussion

### The Relationship between Screen Time and ADHD Risk

In this study, we observed a significant positive correlation between screen time and the risk of ADHD among children. Our study observed a significant positive correlation between screen time and the risk of ADHD among children, consistent with numerous international studies [19]. This association may be attributed to the combined effects of multiple mechanisms. Firstly, the use of screen devices, especially before bedtime, may disrupt children's sleep

**Table 6. Distribution of children at risk of ADHD across different screen time categories.**

| Category of screen time | No. of ADHD risk children | No. of normal children | Total |
|---|---|---|---|
| 0 mins/d | 418 | 3,359 | 3,777 |
| 1~60 mins/d | 5,667 | 29,276 | 34,943 |
| 61~120 mins/d | 660 | 1,804 | 2,464 |
| >120 mins/d | 104 | 206 | 310 |
| Total | 6,849 | 34,645 | 41,494 |

patterns by inhibiting the secretion of melatonin, thereby affecting their attention and emotional regulation abilities [20–22]. Secondly, the fast-paced and highly stimulating content on screens may overstimulate children's attention systems, impacting their cognitive and attentional development [23,24]. Additionally, increased screen time often comes at the expense of physical activity, which has a beneficial effect on improving children's attention and reducing hyperactive behaviors [25,26]. Furthermore, excessive screen time may reduce children's social interactions with peers, and a lack of social skills is associated with the development of ADHD, leading to peer relationship problems and difficulties in school adjustment [27–29]. These findings underscore the importance of reducing children's screen time to preventing ADHD.

## The Relationship between Screen content and ADHD Risk

Educational videos, as a medium for children to acquire knowledge, do not universally exert positive influences [30,31]. Previous research findings on educational videos have been inconsistent. Some studies suggest that educational videos do not significantly increase the risk of attention deficit hyperactivity disorder (ADHD) [32], while others have found that prolonged exposure to educational videos may adversely affect children's attentional systems [33]. This study reveals that as the time spent watching educational videos increases, the risk of ADHD among children rises significantly. This may be attributed to the fact that educational videos often contain a wealth of information with rapid scene changes, which can easily overstimulate children's attentional systems, thereby impairing their self-regulation abilities [34,35]. Additionally, the lack of interactivity in educational videos may deprive children of opportunities to practice and apply knowledge in real-life situations, impacting their social skills and problem-solving abilities [36,37].

Cartoon videos, with their vibrant colors and exaggerated movements, are popular among children, but prolonged viewing may also elevate the risk of ADHD. The fast-paced and highly stimulating content in cartoon videos may excessively activate children's attentional systems [38], leading to decreased attention to the real world and affecting cognitive and emotional development [39]. Moreover, violent or stimulating content in cartoon videos may adversely affect children's mental health [40,41], further increasing the risk of ADHD.

This study finds that, unlike educational and cartoon videos, interactive videos do not show a significant association with the risk of ADHD. Biofeedback therapy, used in attention training for children with ADHD [42], leverages interactive videos to train children's attention. During these sessions, children must adjust their brain electrical activity in real-time while watching interactive videos, thereby enhancing their attention [43,44]. Interactive videos provide children with a greater sense of participation and feedback opportunities [45,46], which may contribute to the lack of significant association between interactive videos and ADHD risk. However, this does not imply that children should increase their use of interactive videos indiscriminately. Dialectically speaking, the point estimate is consistent with other

analyses in the interactive videos group, but the confidence interval is wide, which cannot rule out potential differences. This may be due to insufficient power resulting from a smaller sample size in this stratum. Therefore, it will be necessary to increase the sample size of this group to further validate this result in future studies.

This study possesses notable strengths, primarily reflected in its large sample size (41,494 children) and meticulous data analysis, allowing for an in-depth investigation into the relationship between screen time and ADHD risk, as well as the specific impacts of various screen contents (educational videos, cartoon videos, and interactive videos). However, several limitations merit further attention. Firstly, the cross-sectional design precludes direct causal inference, necessitating future longitudinal research to validate these findings. Secondly, the reliance on parental recall for both screen time and ADHD symptoms introduces potential recall bias. Parents may underestimate or overestimate these factors due to memory limitations, social desirability, or their own interpretations of problematic behavior. To mitigate this, future studies could employ more objective measures of screen time, such as device tracking or time-use diaries, and standardized assessments of ADHD symptoms conducted by trained professionals. Additionally, despite controlling for multiple confounding factors, there may still be unrecognized or unmeasured variables influencing the results, including genetic factors, environmental exposures, or other lifestyle variables. Future research should strive to identify and control for these additional factors. Lastly, a specific concern is the potential for diagnostic bias in parent-reported ADHD symptoms. ADHD is a complex disorder with overlapping symptoms, and parents may not fully understand the diagnostic criteria. To address this, future studies should incorporate clinical assessments by professionals to ensure accurate diagnosis.

## Conclusion

Increased screen time is associated with a higher risk of ADHD, particularly for educational and cartoon videos, while interactive videos show no significant link. To mitigate this risk, parents and educators should implement strategies such as setting time limits, encouraging breaks, and promoting alternative activities. Future research should focus on longitudinal studies and intervention trials to further explore and address this relationship.

## Supporting information

**S1 Data. Minimal_Data_Set_for_ADHD_Screen_Time_Study.**
(XLSX)

## Acknowledgments

The author would like to express his gratitude to the participants of the study, the investigators of Shenzhen Longhua District Maternal and Child Health Hospital and the kindergarten teachers who participated in the survey.

## Author contributions

**Conceptualization:** Jian-Bo Wu, Jian-hui Chen.

**Data curation:** Jian-Bo Wu, Shuang-Yan Qiu, Zhaodi Chen.

**Formal analysis:** Jian-Bo Wu, Jian-hui Chen.

**Funding acquisition:** Jian-Bo Wu, Qiang Zhou, Zhaodi Chen.

**Investigation:** Jian-Bo Wu, Shuang-Yan Qiu, Jingyu Zhang.

**Methodology:** Jian-hui Chen, Zhaodi Chen.

**Project administration:** Jian-Bo Wu, Yanni Yang, Xiaona Yin.

**Resources:** Jian-Bo Wu, Yawei Guo, Zhaodi Chen.

**Software:** Yanni Yang, Jian-hui Chen.

**Supervision:** Wei-Kang Yang, Minghui Meng.

**Validation:** Jian-Bo Wu, Zhaodi Chen.

**Visualization:** Jian-Bo Wu, Jiemin Li, Jian-hui Chen.

**Writing – original draft:** Jian-Bo Wu, Jian-hui Chen, Zhaodi Chen.

**Writing – review & editing:** Jian-Bo Wu, Jian-hui Chen, Zhaodi Chen.

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
