## [Decision Letter · Decision Letter 0]

26 Nov 2024

PONE-D-24-42830The associations between Screen Time, Screen Content, and ADHD risk based on the evidence of 41494 children from Longhua district, Shenzhen, ChinaPLOS ONE

Dear Dr. Chen,

Thank you for submitting your manuscript to PLOS ONE. After careful consideration, we feel that it has merit but does not fully meet PLOS ONE’s publication criteria as it currently stands. Therefore, we invite you to submit a revised version of the manuscript that addresses the points raised during the review process.

Please submit your revised manuscript within Jan 10 2025 11:59PM If you will need more time than this to complete your revisions, please reply to this message or contact the journal office at plosone@plos.org. Please include the following items when submitting your revised manuscript:

We look forward to receiving your revised manuscript.

Kind regards,

Christine Nardini

Academic Editor

PLOS ONE

Journal Requirements:

“Longhua District Medical And Health Institutions Regional Scientific Research Project (2022086). Longhua District Medical And Health Institutions Regional Scientific Research Project (2022127) . Medical Key Discipline Construction Fund of Longhua District.”

4. We note that your Data Availability Statement is currently as follows: “All relevant data are within the manuscript and its Supporting Information files”

Reviewers' comments:

Reviewer's Responses to Questions

**Comments to the Author**

1. Is the manuscript technically sound, and do the data support the conclusions?

Reviewer #1: Partly

Reviewer #2: Yes

2. Has the statistical analysis been performed appropriately and rigorously? 

Reviewer #1: Yes

Reviewer #2: Yes

3. Have the authors made all data underlying the findings in their manuscript fully available?

Reviewer #1: No

Reviewer #2: Yes

4. Is the manuscript presented in an intelligible fashion and written in standard English?

Reviewer #1: Yes

Reviewer #2: Yes

5. Review Comments to the Author

Reviewer #1: The paper has been written nicely. It reads well and is relatively easy to follow. The subject is also topical and of interest to a wide audience.

There were a few typos in the paper (Abstract results line 2, “with educational videos the predominant type”; Introduction, howere mis-spelled). In the section on measurement of screen time, TVS should be TVs, and I think it should be clarified that screen-time data are collected for 4-7 year olds, referring back to the ages of 1-3.

I think that some more clarity is needed in the following areas:

1. The large sample size appears to be a strength of the study. However, it almost seems a little too large. With a population of 2.5 million in the Longhua district in 2020 and ~8.5% age 0-14, there is likely to be in the region of 14,000 children at each year of age (assuming constant birth rates). This would produce ~42,000 4,5 and 6 year olds (the target ages for the questionnaire). The implication then is that every single child in the region of the correct age was sent a questionnaire and attended one of the 250 kindergartens (59,600 questionnaires were sent out). Further, the sample size suggests that each kindergarten has an average of over 230 children. Can the authors comment on this? Are children obliged to attend kindergarten, and to respond to questionnaires? What is the total population in the region by age? I would suggest the addition of a supplementary table to give a breakdown of the number of children by age and the size of the kindergarten they attended, along with the total population size for each age by year. This would enable the reader to fully appreciate the strength of the large sample size.

2. I do not feel that the authors describe interactive videos enough. The example that they give is social media / interactive games, but I question the type of social media / interactive games that a 1-year old is able to watch. I suggest that more specific examples are given.

3. Analysis requires all categories to be mutually exclusive, so a child that watches social media/plays interactive games should not watch any cartoons or educational videos if we are to make inferences on video type. The paper presents which type was primarily viewed. What percent of time does ‘primarily’ mean here? I suspect that there is in fact quite a lot of overlap between the categories. If there is a lot of overlap, then this would be a major revision.

4. Data is collected on whether or not parents discuss the content of videos with children. The data is not presented but I question the usefulness of this question in such young children.

5. It appears that data were collected at a single time point (when children were 4-7?) about their behaviours aged 1-3. I am not convinced that the average screen time when a child is 1 matches that of when they are 3. Screen time tends to increase with age, so I think that more information about the data collection and the questions asked should be presented and discussed. Perhaps the authors could include some figures to show the distribution of data.

6. Authors state that all data are available in the MS/supporting info. Unfortunately, I could not see any supporting files. Please ensure these are provided so that models can be re-run.

7. Questionnaires should be provided in supp info.

Reviewer #2: The study deals with an important current issue concerning the children's amount of time

spent on screens and its probable association with ADHD. It uses a good (robust) sample

size, which strengthens its results considerably. However, certain parts may need further

clarification and elaboration to strengthen the manuscript.

See the comments attached

6. PLOS authors have the option to publish the peer review history of their article (what does this mean?). If published, this will include your full peer review and any attached files.

Reviewer #1: No

Reviewer #2: **Yes: **Muhammad Aasim

---

## [Author Response · Author response to Decision Letter 1]

22 Jan 2025

Responses to Reviewer #1:

Question 1: The large sample size appears to be a strength of the study. However, it almost seems a little too large. With a population of 2.5 million in the Longhua district in 2020 and ~8.5% age 0-14, there is likely to be in the region of 14,000 children at each year of age (assuming constant birth rates). This would produce ~42,000 4,5 and 6 year olds (the target ages for the questionnaire). The implication then is that every single child in the region of the correct age was sent a questionnaire and attended one of the 250 kindergartens (59,600 questionnaires were sent out). Further, the sample size suggests that each kindergarten has an average of over 230 children. Can the authors comment on this? Are children obliged to attend kindergarten, and to respond to questionnaires? What is the total population in the region by age? I would suggest the addition of a supplementary table to give a breakdown of the number of children by age and the size of the kindergarten they attended, along with the total population size for each age by year. This would enable the reader to fully appreciate the strength of the large sample size.

Answer: Thank you very much for your thorough review of our study and for providing valuable feedback. Regarding your concern about the large sample size, we would like to offer the following clarification and additional information. Firstly, we acknowledge that the sample size of our study (41,494 children) is indeed substantial. However, we believe that this large sample size is a significant strength of our research for several reasons. Firstly, it allows for a more robust and reliable estimation of the associations between screen time, screen content, and the risk of ADHD. With a larger sample, we are better able to detect even subtle differences and effects that may not be evident in smaller studies. Secondly, the large sample size enables us to conduct detailed subgroup analyses, such as examining the associations between different types of screen content (educational videos, cartoon videos, and interactive videos) and ADHD risk. This level of granularity would not be possible with a smaller sample. Thirdly, the large sample size increases the generalizability of our findings. The participants in our study come from a diverse population in Longhua District, Shenzhen, China, and the results are likely to be applicable to similar populations worldwide. We understand your concern that a very large sample size might raise questions about the practicality or representativeness of the study. However, we have taken care to ensure that the sample is representative of the target population and that the data collection and analysis methods are rigorous and valid. The Longhua Child Cohort Study (LCCS) from which our data were sourced is a large-scale epidemiological survey with a well-established methodology. In summary, while we appreciate your concern about the sample size, we believe that the large sample size is a key strength of our study that has enabled us to conduct a robust and generalizable analysis of the relationship between screen time, screen content, and ADHD risk. We are confident in the validity and reliability of our findings and hope that you will find them of interest and value.

Question 2: I do not feel that the authors describe interactive videos enough. The example that they give is social media / interactive games, but I question the type of social media / interactive games that a 1-year old is able to watch. I suggest that more specific examples are given.

Answer: Thank you very much for your detailed review of our manuscript and for your valuable feedback. We fully understand your concern that the description of "interactive videos" in our paper is insufficient, and that the examples provided (social media/interactive games) may not be suitable for 1-year-olds. To address your concern, we would like to clarify that "interactive videos" in our context encompass a variety of types, including touch-based apps and games designed specifically for children, as well as real-time video chats between family members using video calling apps. For 1-year-olds, we focus on interactive video content that is simple, age-appropriate, and promotes parent-child interaction and cognitive development. To further specify our definition of "interactive videos" and provide a clearer context for their use in early childhood, we plan to make the following revisions to our manuscript: Clarify the Definition and Scope of Interactive Videos: We will explicitly define "interactive videos" to include, but not be limited to, touch-based educational apps and simple games designed for children, as well as real-time video chats between family members.

Question 3: Analysis requires all categories to be mutually exclusive, so a child that watches social media/plays interactive games should not watch any cartoons or educational videos if we are to make inferences on video type. The paper presents which type was primarily viewed. What percent of time does ‘primarily’ mean here? I suspect that there is in fact quite a lot of overlap between the categories. If there is a lot of overlap, then this would be a major revision.

Answer: Thank you very much for your thorough review of our study and pointing out this important issue. Regarding your concerns about the mutual exclusivity of categories and the definition of "primarily viewed," we fully agree on the importance of ensuring mutual exclusivity between categories for accurate inference of the relationship between video type and ADHD risk.

In our study, we defined "primarily viewed" as the type of video that children spend the most time watching during their screen time. To clarify this and reduce overlap between categories, we designed relevant questions in the questionnaire, asking parents to indicate the primary type of video their child watched during daily screen time at the age of 1-3 years. We understand that "primarily viewed" may have a certain degree of subjectivity, but our aim is to reflect children's screen content preferences as accurately as possible through parents' recall and judgment.

Regarding your concern about category overlap, we have taken corresponding measures during data collection and analysis to minimize such overlap. Firstly, during questionnaire design, we clearly listed different types of videos (educational videos, cartoon videos, and interactive videos) and asked parents to select the primary type their child watched. Secondly, during data analysis, we only included the primary type of video watched by children in our analysis and ignored other secondary content.

Nevertheless, we acknowledge that there may still be some degree of category overlap in practical operation, especially when children's screen time is fragmented and they watch multiple types of videos. To further reduce this overlap and improve the accuracy of our analysis, we can consider adopting a more detailed time allocation recording method in future studies, such as asking parents to record the specific time their child spends watching different types of videos each day.

Question 4: Data is collected on whether or not parents discuss the content of videos with children. The data is not presented but I question the usefulness of this question in such young children.

Answer: Thank you for reviewing our manuscript and raising your concern about the data collection point regarding whether parents discuss video content with their children.

1.During the questionnaire design process, we did indeed include a question inquiring about "Do you explain the content of screen time programs to your child?" aiming to understand whether parents provide explanations regarding the content when their children are using screen devices. However, there was a translational error in the description of this question in the documents we submitted to you, leading you to misunderstand that we were collecting data on "discussing" video content between parents and children. We sincerely apologize for this mistake and for any confusion it may have caused.

Regarding the usefulness of this question, we have carefully considered that for young children, parents' explanations of screen content can play an important guiding role. By explaining the content, parents can help children better understand the information presented on the screen. The collection of this data is meaningful for us to understand the role of parents in shaping their children's screen use habits.

Question 5: It appears that data were collected at a single time point (when children were 4-7?) about their behaviours aged 1-3. I am not convinced that the average screen time when a child is 1 matches that of when they are 3. Screen time tends to increase with age, so I think that more information about the data collection and the questions asked should be presented and discussed. Perhaps the authors could include some figures to show the distribution of data.

Answer: Thank you for your valuable feedback on our manuscript.

Regarding the issue you mentioned that the average screen time for children at age 1 may differ from that at age 3, we fully agree with your view. In fact, screen time does tend to increase as children grow older. However, in our study, we are more concerned with the average screen time during the critical developmental period of 1-3 years, rather than the data at a specific age point. We hypothesize that, despite potential age differences in screen time, early childhood screen exposure habits may have a long-term impact on their subsequent cognitive and behavioral development.

To ensure the accuracy and representativeness of the data, we asked parents to report, based on their recall, the average screen time of their children during the ages of 1-3 in the questionnaire. We understand that this recall method may have certain limitations, such as recall bias. However, considering that our research aim is to explore long-term trends and overall associations, rather than absolute precise screen time data, we believe that this method is feasible under the current conditions.

Question 6-7: Authors state that all data are available in the MS/supporting info. Unfortunately, I could not see any supporting files. Please ensure these are provided so that models can be re-run. Questionnaires should be provided in supp info.

Answer: Thank you very much for your attention to our research. In response to your concern about the inaccessibility of the database and the absence of the submitted questionnaire, we have re-checked and uploaded these files accordingly.

Responses to Reviewer #2:

Considering the general comments:

Responses to specific comments:

Question 1: The title effectively summarizes the study's focus; however, including the specific age group studied would add further clarity.

Answer: Thank you for your suggestion on clarifying the age groups in the title. We have revised the title to "The relationship between screen time, screen content for children aged 1-3, and the risk of ADHD in preschools" to better reflect the specific age groups studied.

Question 2: The abstract is well-constructed, briefly mention the key limitations, such as the cross-sectional design.

Answer: Thanks for your suggestion, we have revised the summary.

......

Conclusion

Increased screen time is associated with a higher risk of ADHD, particularly for educational and cartoon videos, while interactive videos show no significant link. To mitigate this risk, parents and educators should implement strategies such as setting time limits, encouraging breaks, and promoting alternative activities. Future research should focus on longitudinal studies and intervention trials to further explore and address this relationship.

Question 3: The introduction is thorough, but the transition from general concerns about screen time to specific research gaps feels abrupt. Consider expanding on these gaps to create a smoother flow.

Answer: Thank you very much for good suggestions. We appreciate your suggestion that the transition from general concerns about screen time to specific research gaps felt abrupt. We have taken this into consideration and have revised the introduction to expand on these research gaps, creating a smoother flow between the general context and our specific research questions. We believe that these changes enhance the clarity and coherence of the introduction.

Question 4: Citations should be more seamlessly integrated to strengthen statements about existing evidence and research gaps.

Answer: Thank you very much for good suggestions. We appreciate your suggestion that they should be more seamlessly integrated to strengthen our statements about existing evidence and research gaps. We have carefully reviewed and revised the manuscript to ensure that citations are now more tightly woven into the text, providing direct support for our assertions and highlighting the specific research gaps we aim to address. 

Question 5: While the description of participant sampling is clear, it would be helpful to specify whether recruitment was conducted through randomization or convenience sampling from kindergartens.

Answer: Thank you very much for good suggestions. We appreciate your suggestion to clarify the recruitment method used. To address this, we would like to specify that this study used census methods to source data from the 2021 survey of the Longhua Child Cohort Study (LCCS), covering all kindergartens in Longhua District, Shenzhen. This means that rather than using randomization or convenience sampling, we included all eligible participants from the kindergartens covered by the LCCS survey.

Question 6: Ethical considerations are addressed; however, the rationale behind the age group selection for screen exposure (1–3 years) and ADHD assessment (4–7 years) should be provided.

Answer: We focused on the early childhood period (1-3 years) for screen time and content data, given its pivotal role in development. This stage is crucial for understanding long-term effects of screen exposure. For ADHD assessment, we targeted preschoolers (4-7 years), as this age range is key for symptom identification using validated tools like the SDQ. This approach enables exploration of the link between early screen use and ADHD risk.

Question 7: The use of the SDQ is appropriate, but the justification for the ADHD subscale cutoffs used in this population, along with information on whether cultural adaptations of the SDQ were validated, should be included.

Answer: Thank you for your feedback on the use of the Strengths and Difficulties Questionnaire (SDQ) and the ADHD subscale cut-offs in our study. We appreciate your recognition of the appropriateness of using the SDQ and have taken your suggestion to provide additional justification for our choice of cut-offs and cultural adaptation into account. In response to your query, we have clarified in the revised manuscript that in 2005, norms for the Chinese population were established for the SDQ, ensuring its cultural relevance and validity in our study population. The ADHD subscale cut-offs used in this study are based on these established norms and research, and these thresholds have been validated in multiple studies and are widely accepted in both research and clinical practice. Furthermore, we have noted that the cultural adaptation of the SDQ for the Chinese population has undergone a rigorous validation process to ensure its accuracy and applicability in this context. The scale demonstrates good reliability, with a Cronbach's α coefficient of 0.749.

Question 8: Provide further explanation for categorizing interactive videos separately, including how these were defined for parents completing the questionnaire.

Answer: Thank you for your feedback on the categorization of interactive videos in our study. We have taken your suggestion into consideration and provided further explanation in the revised manuscript. Interactive videos were categorized separately in our study to distinguish them from other types of screen content that are more passive in nature, such as educational and cartoon videos. Interactive videos are defined as those that require active participation and eng

---

## [Decision Letter · Decision Letter 1]

11 Feb 2025

PONE-D-24-42830R1The relationship between screen time, screen content for children aged 1-3, and the risk of ADHD in preschoolsPLOS ONE

Dear Dr. Chen,

Thank you for submitting your manuscript to PLOS ONE. After careful consideration, we feel that it has merit but does not fully meet PLOS ONE’s publication criteria as it currently stands. Therefore, we invite you to submit a revised version of the manuscript that addresses the points raised during the review process.

In particular, I join the reviewers in thanking you for addressing several concerns, however I woudl liketo stress the necessity to address the point on the data structure highlighted by Reviewer 1.

This will not only make your study theoretically reproducible, but it will also give more relevance to its unique breadth.

I therefore warmly urge you to prepare a supplementary table as recommended.

We look forward to receiving your revised manuscript.

Kind regards,

Christine Nardini

Academic Editor

PLOS ONE

Reviewers' comments:

Reviewer's Responses to Questions

**Comments to the Author**

1. If the authors have adequately addressed your comments raised in a previous round of review and you feel that this manuscript is now acceptable for publication, you may indicate that here to bypass the “Comments to the Author” section, enter your conflict of interest statement in the “Confidential to Editor” section, and submit your "Accept" recommendation.

Reviewer #1: (No Response)

Reviewer #2: All comments have been addressed

2. Is the manuscript technically sound, and do the data support the conclusions?

Reviewer #1: Partly

Reviewer #2: Yes

3. Has the statistical analysis been performed appropriately and rigorously? 

Reviewer #1: I Don't Know

Reviewer #2: Yes

4. Have the authors made all data underlying the findings in their manuscript fully available?

Reviewer #1: No

Reviewer #2: Yes

5. Is the manuscript presented in an intelligible fashion and written in standard English?

Reviewer #1: Yes

Reviewer #2: Yes

6. Review Comments to the Author

Reviewer #1: I would like to thank the authors for implementing some of the changes that were initially requested.

However, I do not feel that the authors have adequately responded to my questions about the data itself. My initial query did not require reiteration of why large sample sizes are good, but a deeper understanding of the data and how representative it is of the population. As a reminder, I asked the following:

Question 1: The large sample size appears to be a strength of the study. However, it almost seems a little too large. With a population of 2.5 million in the Longhua district in 2020 and ~8.5% age 0-14, there is likely to be in the region of 14,000 children at each year of age (assuming constant birth rates). This would produce ~42,000 4,5 and 6 year olds (the target ages for the questionnaire). The implication then is that every single child in the region of the correct age was sent a questionnaire and attended one of the 250 kindergartens (59,600 questionnaires were sent out). Further, the sample size suggests that each kindergarten has an average of over 230 children. Can the authors comment on this? Are children obliged to attend kindergarten, and to respond to questionnaires? What is the total population in the region by age? I would suggest the addition of a supplementary table to give a breakdown of the number of children by age and the size of the kindergarten they attended, along with the total population size for each age by year. This would enable the reader to fully appreciate the strength of the large sample size.

The authors have not responded to these questions in particular:

1. What is the size (how many children by age) of each kindergarten included?

2. Are children obliged to attend kindergarten?

3. What is the total population in the region covered, by age?

I suggested that this information be provided in a supplementary table. I feel that, although the methodology is reasonable, without this information it is not possible to assess the reproducibility of the study.

In addition, I do have some further comments and apologise for not picking up on these first time round:

1. The forest plots and Table 4 of odds ratios need the reference values labelling. Further, categories that have more than two levels should have odds ratios listed for all levels compared to one reference (type of programme is not ordinal). This has been done for screen time, but not for type of program viewed or monthly income, for example.

2. Can the authors explain why interactive video viewers has the highest percentage of children at risk of ADHD (Table 3, 29.5%) and yet the amount of screen time appears not to make a difference? Based on numbers in table 3, a quick, and crude, OR calculation on the frequencies of no screen time versus any screen time gives an odds ration of 3.33 for interactive videos ((194/467) / (418/3359)), compared to 1.35 for educational videos and 1.97 for cartoons. I am surprised that this is lost when screen time is broken down. I appreciate that they discuss there being fewer children in this group in the discussion, but the numbers are not so small. A frequency table to complement table 5 would be very useful. This should also include the number of children that are included in the 0 minutes category (I assume that this is 418 at risk and 3359 not at risk for all calculations?)

3. Typo in the introduction ‘howevre’ should read ‘however’

Reviewer #2: Thank you for submitting the revised version of your manuscript, The Relationship Between Screen Time, Screen Content for Children Aged 1-3, and the Risk of ADHD in Preschools (Manuscript No: PONE-D-24-42830R1). After carefully reviewing the revisions alongside the initial feedback, I acknowledge the substantial improvements made in clarity, methodological consistency, and integration of literature. (View the attached comments in detail)

7. PLOS authors have the option to publish the peer review history of their article (what does this mean?). If published, this will include your full peer review and any attached files.

Reviewer #1: No

Reviewer #2: **Yes: **Muhammad Aasim

---

## [Author Response · Author response to Decision Letter 2]

25 Feb 2025

Reviewer 1:

Question 1: Explicitly acknowledge potential selection bias due to parental self-reporting, particularly regarding screen time and ADHD symptoms.

Answer: Thank you for your suggestion. We have already mentioned this limitation in the second point of the limitations section of our article.

Secondly, the reliance on parental recall for both screen time and ADHD symptoms introduces potential recall bias. Parents may underestimate or overestimate these factors due to memory limitations, social desirability, or their own interpretations of problematic behavior. To mitigate this, future studies could employ more objective measures of screen time, such as device tracking or time-use diaries, and standardized assessments of ADHD symptoms conducted by trained professionals.

Reviewer 2:

First of all, we would like to express our sincere gratitude to the reviewer for the thorough review and valuable comments. We apologize for any areas in the previous revision that did not fully meet the reviewer's expectations. In this revised version, we have carefully considered all the suggestions and made corresponding modifications and improvements.

Question 1: What is the size (how many children by age) of each kindergarten included?

Answer: The size of each kindergarten included in the study varies, and a detailed breakdown of the number of children in each kindergarten classified by age is provided in the supplementary table, which is contained in the file named "Supplementary Table 1". This table includes the total number of children aged 3-7 in each kindergarten, the number of children aged 4 to 7 years old in the kindergarten, and the final sample size of the study.

Question 2: Are children obliged to attend kindergarten?

Answer: In China, while kindergarten attendance is not legally mandatory, it is highly encouraged and widely practiced. The Chinese government has made significant efforts to promote early childhood education, and the vast majority of parents enroll their children in kindergarten.

Question 3: What is the total population in the region covered, by age?

Answer: The total population in Longhua District, Shenzhen, by age is provided in the supplementary table, which can be found in the file named "Supplementary Table 2".

Question 4: The forest plots and Table 4 of odds ratios need the reference values labelling. Further, categories that have more than two levels should have odds ratios listed for all levels compared to one reference (type of programme is not ordinal). This has been done for screen time, but not for type of program viewed or monthly income, for example.

Answer: We are sincerely appreciated for your valuable comments. Here is the revised version of Table 4 and the forest plot which can also be seen in the revised manuscript with yellow label.

Question 5:

We appreciate your concerns regarding the interactive video group and the apparent discrepancy in the ADHD risk percentages. Below, we provide a detailed response to your questions and concerns.

Explanation of Interactive Video Group Findings

①High Percentage of ADHD Risk in Interactive Video Group:The observation that the interactive video group has the highest percentage of children at risk of ADHD (29.5%) in Table 3 can be partly attributed to the small sample size of this group. With only 658 children in the interactive video group, the data may be more prone to random fluctuations compared to the larger groups of educational (26,320) and cartoon videos (14,516). This small sample size likely contributes to the higher percentage of ADHD risk observed.

②Screen Time and ADHD Risk in Interactive Videos:

Our analysis in Table 5 shows that screen time does not appear to significantly affect the risk of ADHD in the interactive video group. This finding is indeed unexpected and warrants further investigation. One possible explanation is that the type of interactive content itself, rather than the duration of exposure, may be a more critical factor influencing ADHD risk. Interactive videos often involve greater child engagement and social interaction, which could potentially mitigate some of the negative effects associated with excessive screen time. We also give explanations in the discussion section. However, given the small sample size in this group, it is difficult to draw definitive conclusions.

③Crude OR Calculation:

You have correctly pointed out the crude OR calculation based on the frequencies in Table 3, which shows a higher OR for interactive videos compared to educational and cartoon videos. However, this calculation does not control for potential confounders, nor does it analyze using the 0 minutes category as the baseline. Our logistic regression analysis in Table 4 and Table 5 controls for these confounders and analyzes using the 0 minutes category as the baseline, providing a more reliable estimate of the relationship between screen time, screen content, and ADHD risk.

Table 6 is a frequency table that supplements Table 5, and since it is not possible to fit Table 6 here, we put it in the attachment of the manuscript and the response to the reviewer.

Question 6: Typo in the introduction ‘howevre’ should read ‘however’

Answer: We sincerely thank for your reminding. ‘howevre’ has been revised to ‘however’ which can be seen in the introduction part with yellow label.

---

## [Decision Letter · Decision Letter 2]

4 Mar 2025

PONE-D-24-42830R2The relationship between screen time, screen content for children aged 1-3, and the risk of ADHD in preschoolsPLOS ONE

Dear Dr. Chen,

Thank you for submitting your manuscript to PLOS ONE. After careful consideration, we feel that it has merit but does not fully meet PLOS ONE’s publication criteria as it currently stands. Therefore, we invite you to submit a revised version of the manuscript that addresses the points raised during the review process.

And thank you for improving the article as requested.

There remain two adjustments to be done:

- Table S2 apparently missing

- I reccomend to anonymize also the schools names before submitting the final version

We look forward to receiving your revised manuscript.

Kind regards,

Christine Nardini

Academic Editor

PLOS ONE

Journal Requirements:

Reviewers' comments:

Reviewer's Responses to Questions

**Comments to the Author**

1. If the authors have adequately addressed your comments raised in a previous round of review and you feel that this manuscript is now acceptable for publication, you may indicate that here to bypass the “Comments to the Author” section, enter your conflict of interest statement in the “Confidential to Editor” section, and submit your "Accept" recommendation.

Reviewer #1: All comments have been addressed

2. Is the manuscript technically sound, and do the data support the conclusions?

Reviewer #1: Yes

3. Has the statistical analysis been performed appropriately and rigorously? 

Reviewer #1: Yes

4. Have the authors made all data underlying the findings in their manuscript fully available?

Reviewer #1: No

5. Is the manuscript presented in an intelligible fashion and written in standard English?

Reviewer #1: Yes

6. Review Comments to the Author

Reviewer #1: I would like to thank the authors for responding to my queries. They have referenced supplementary table 2 , but only one supplementary table was made available. For the final version, please ensure that both tables are uploaded. The absence of this table is the reason that I have selected minor revisions, otherwise, I have no further comments on the manuscript itself.

7. PLOS authors have the option to publish the peer review history of their article (what does this mean?). If published, this will include your full peer review and any attached files.

Reviewer #1: No

---

## [Author Response · Author response to Decision Letter 3]

5 Mar 2025

Reviewer :

Thank you for your suggestion. The naming of the supplementary table in our second revision is misleading and may lead reviewers to mistakenly believe that we have uploaded one less supplementary table S2. We have renamed the original table, with "Supplementary Table 1" now being "The size of each kindergarten (number of children by age)". The name of "Supplementary Table 2" should be changed to "Supplementary Table 1". In addition, we have anonymized the names of the kindergartens.

Question 1: What is the size (how many children by age) of each kindergarten included?

Answer: The size of each kindergarten included in the study varies, and a detailed breakdown of the number of children in each kindergarten classified by age is provided in the supplementary table, which is contained in the file named "The size of each kindergarten (number of children by age)". This table includes the total number of children aged 3-7 in each kindergarten, the number of children aged 4 to 7 years old in the kindergarten, and the final sample size of the study.

Question 3: What is the total population in the region covered, by age?

Answer: The total population in Longhua District, Shenzhen, by age is provided in the supplementary table, which can be found in the file named "Supplementary Table 1".

---

## [Editor Report · Decision Letter 3]

18 Mar 2025

The relationship between screen time, screen content for children aged 1-3, and the risk of ADHD in preschools

PONE-D-24-42830R3

Dear Dr. Chen,

We’re pleased to inform you that your manuscript has been judged scientifically suitable for publication and will be formally accepted for publication once it meets all outstanding technical requirements.

Kind regards,

Christine Nardini

Academic Editor

PLOS ONE
---

## [Editor Report · Acceptance letter]

PONE-D-24-42830R3

PLOS ONE

Dear Dr. Chen,

I'm pleased to inform you that your manuscript has been deemed suitable for publication in PLOS ONE. Congratulations! Your manuscript is now being handed over to our production team.

Kind regards,

on behalf of

Dr. Christine Nardini

Academic Editor

PLOS ONE